# Effect of Pre-Germination Temperature Regime on Pollen Germination and Fruit Set in Pear, *Pyrus bretschneiderilia*

**Limin Liu** [1,2,3], **Ziyan Liu** [1,2,3], **Hu Han** [1,2,3], **Baiyu Qiao** [1,2,3], **Yangfan Li** [1,2,3], **Xiongkui He** [1,2,3,*]
and **Yajia Liu** [1,2,3,*]

1    College of Science, China Agricultural University, Beijing 100193, China; liulimsy2882@163.com (L.L.);
     s20213102004@cau.edu.cn (Z.L.); hh2931434722@163.com (H.H.); byq992670374@163.com (B.Q.);
     s20203101935@cau.edu.cn (Y.L.)
2    Centre for Chemicals Application Technology, China Agricultural University, Beijing 100193, China
3    College of Agricultural Unmanned System, China Agricultural University, Beijing 100193, China
*    Correspondence: xiongkui@cau.edu.cn (X.H.); liuyajia@cau.edu.cn (Y.L.)

**Abstract:** Extensive research has been conducted on the effects of pollen activation temperature, cultivation temperature, and ambient temperature on germination rates and fruit sets. Yet, the influence of the temperature (pre-germination temperature) at which the pollen suspension is prepared within the storage container (tank) remains unexplored. In this study, we initially established the optimal time for pollen activation. Subsequently, pollen suspensions at varying temperatures were prepared, and their germination rates were analyzed using microscopy. Pollen suspensions of different temperatures were then sprayed onto Fojianxi pear flowers, and parameters such as flower fruit set rate, inflorescence fruit set rate, fruit longitudinal dimension, fruit transverse dimension, and fruit shape index were evaluated. The findings revealed that the optimal pollen activation duration was 6 h. A pollen suspension temperature of 30 °C resulted in the highest germination rate (72.06 ± 6.87%). However, a suspension temperature of 25 °C achieved the highest flower fruit set rate (57.29 ± 5.58%) and inflorescence fruit set rate (87.50 ± 4.84%). The fruit longitudinal dimension (68.12 ± 3.94 mm), transverse dimension (73.77 ± 4.04 mm), and fruit shape index (108.42 ± 6.02%) were most favorable at a suspension temperature of 15 °C. Our study concludes that the pollen in lower pollen suspension temperatures (15~25 °C) has higher germination energy and have higher stigmatic capacity. Thus, we advocate for a pollen suspension temperature range of 15~25 °C when employing liquid spray pollination techniques with Xuehuali (*Pyrus bretschneiderilia*) pollen.

**Keywords:** liquid spray pollination; fruit set; fruit longitudinal stem; fruit transverse stem; germination energy

## 1. Introduction

Pollination stands as a pivotal phase in the sexual reproduction of seed plants [1]. The vitality of pollen and the receptivity of the stigma play direct roles in determining pollination outcomes [2]. Achieving pollination during the period of maximal pollen germination and strongest stigma receptivity is a critical strategy to enhance the fruit set rate in fruit trees [3]. Factors intrinsic to plants, such as enzymes, protein regulations, and mineral elements, as well as extrinsic factors such as temperature and humidity, influence the pollinability of the stigma and pollen vitality. Poor pollination and low pollen germination rate can lead to malformed or small fruits [3].

However, the decline in natural pollinators due to various reasons has necessitated a shift towards artificial pollination [4–7]. This artificial pollination can be executed through manual means, mechanical methods, or other techniques [8]. Mechanical pollination, because of its efficiency, cost-effectiveness, and superior outcomes, is gradually superseding other methods [9]. Concurrently, the technique of liquid spray pollination has gained

traction in mechanical pollination due to its straightforwardness, rapidity, and compatibility with current spraying apparatus [9,10].

Flower buds were collected 1 to 2 days before the pear flower bloomed. The petals were removed from the collected flowers and subsequently the flowers were held in both hands so that they rubbed against each other. This caused the anthers to fall off the flowers, which were then spread evenly on smooth paper. The pollen was released after 24~48 h of shade drying in a room at 23~25 °C. After drying the pollen was placed in dry containers and stored at low temperatures. When pollination took place, the pollen was taken out and activated [11]. Temperature emerges as a decisive factor influencing pollen germination across all pollination techniques [12,13]. Especially in artificial pollination, the storage temperature of pollen holds significance. Typically, pollen can be preserved between −40 °C and −18 °C for several years [14,15]. For liquid spray pollination, an activation process is essential for the pollen [16,17]. Existing research indicates the optimal activation temperature for most pear pollen to be 25 °C [18,19]. The environmental temperature during liquid spray pollination cannot be overlooked [20,21]. Extensive studies employing liquid culture methods to incubate pollen under varied temperatures for extended durations (>6 h) revealed that both excessively high and low temperatures impede optimal pollen germination and pollen tube elongation [22,23]. Xuehuali (*Pyrus bretschneiderilia*) pollen has a higher pollen germination rate [24]. Additionally, the compatibility of the pollen was higher [25]. Therefore, *Pyrus bretschneiderilia* was selected as the father parent in this paper. Meanwhile, the optimal pollen activation and in vitro culture temperature of *Pyrus bretschneiderilia* is 25 °C.

Some researchers have defined the duration from when pollen suspension is prepared until it is sprayed onto the stigma as its "storage time" [26]. Preliminary findings suggest that when this storage duration is below 20 min, the pollen germination rate aligns with that of directly sprayed pollen. This underscores the idea that the longevity of pollen suspension within the storage container (tank) might alter pollen germination rates. Consequently, one might postulate that the temperature of the pollen suspension inside the container could influence both germination rates and the outcomes of liquid spray pollination.

Despite these insights, there remains a gap in the literature regarding the impact of pollen suspension (in the tank) temperature on pollen germination and fruit set efficacy. To address this, our study first determined the optimal activation duration for the target pollen. Subsequently, we prepared pollen suspensions at various temperatures, maintaining them for specific intervals to gauge their germination rates. Concurrently, we applied these differently-tempered pollen suspensions to pear trees, quantifying parameters such as fruit set rate of flower and inflorescence, fruit longitudinal stem, fruit transverse stem, and fruit morphological indices. Our findings aim to guide the refinement of liquid spray pollination techniques by suggesting appropriate temperatures for pollen suspension preparation.

## 2. Materials and Methods

### 2.1. Configuration of Pollen Suspension and Determination of Pollen Germination Rate

Pollen from *Pyrus bretschneiderilia*, harvested in Sanmenxia (34.6203° N, 111.6973° E), Henan Province, China, was collected on 22 March 2021. Subsequent to the drying process, it was stored at −18 °C in a refrigerator commencing from 23 March 2021. The methodology and composition for the pollen solution were aligned with the parameters set by Liu et al. [26]. Table 1 delineates the component content for the 5 L pollen solution as an exemplification.

The pollen and its resultant solution were quantified, adhering to the requisite pollen grain concentration of 0.8 g·L$^{-1}$. The germination rate of the pollen grains, subjected to a liquid culture, was microscopically observed (Model SZM45B1, Shunyu Technology Co., Ltd., Changzhou, China). Following the configuration or spraying of the pollen suspension, a 100 μL aliquot was transferred to a concave slide (dimensions 28 mm × 76 mm). Given the requisite humidity for pollen germination, three pieces of moistened filter paper were placed within culture dishes (10 cm in diameter). Concurrently, the concave slides were

situated atop the moistened filter paper. The culture dishes were then stationed within an artificial climate box (Model RDN500, Yanghui Technology Co., Ltd., Ningbo, China), maintaining a relative humidity of 95% and a temperature of 25 °C. The pollen grains were incubated in the artificial climate box for 3 h under dark conditions. After incubation, the pollen grains within the concave slides were observed and photographed microscopically. The imagery was captured using a Xiaomi 12X mobile phone (Xiaomi Technology Co., Ltd., Beijing, China) affixed to the microscope eyepiece. Pollen grains exhibiting a pollen tube length exceeding their diameter were categorized as germinated pollen (Figure 1a), while those in Figure 1b were identified as non-germinated pollen grains. The pollen germination was calculated based on the number of total and germinated pollen grains (Equation (1)). Three fields of view were randomly selected from each concave slide for observation.

**Table 1.** Composition of pear pollen solution. The table contains the composition, content, and mass (volume) of each component in the pollen solution.

| Ingredients | Content (%) | Volume or Mass |
|---|---|---|
| Demineralized water | - | Fixed volume to 5 L |
| Sucrose | 10.00 | 0.50 kg |
| Xanthan gum | 0.02 | 1.00 g |
| Calcium gluconate | 0.05 | 2.50 g |
| Boric acid | 0.01 | 0.50 g |

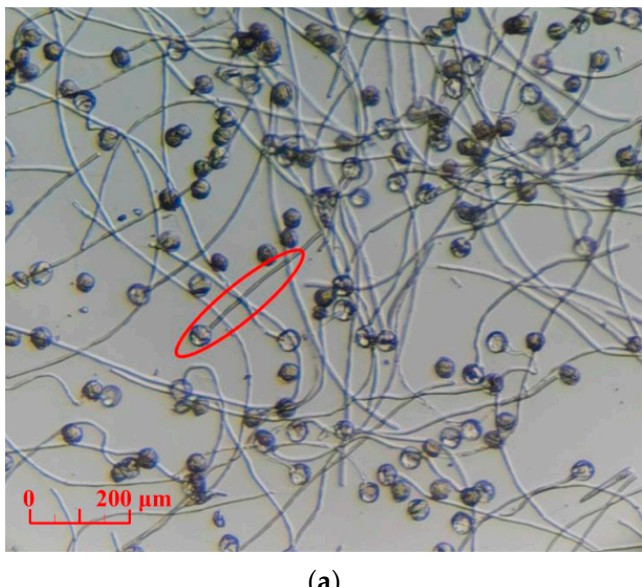

(**a**)

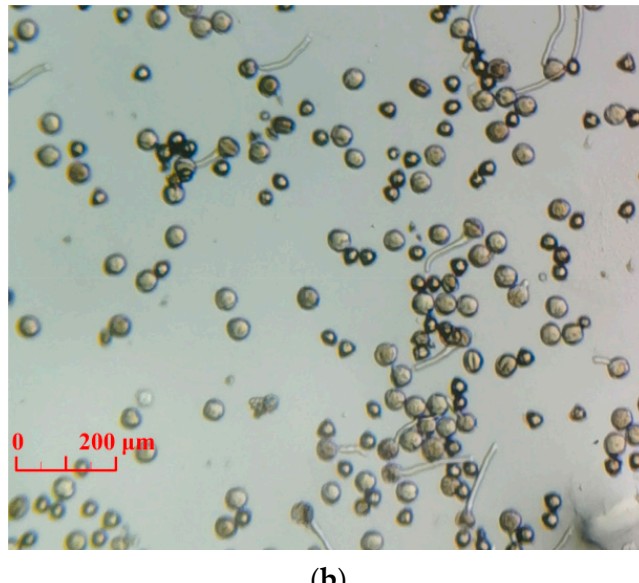

(**b**)

**Figure 1.** Germinated and non-germinated pollen grains under 50× magnification. The germinated pollen was marked by a red oval, which was characterized by the length of the pollen tube exhibiting significantly greater lengths than the pollen grain diameters. (**b**) was non-germinated pollen, which is characterized by no pollen tube. (**a**) Germinated pollen grains. (**b**) Non-germinated pollen grains.

### 2.2. Experiments on the Effect of Incubation Time on the Pollen Germination Rate

Prior to the preparation of the pollen suspension, pollen grains must undergo an activation process. According to pertinent literature, the optimal temperature for pollen activation is 25 °C, with a relative humidity of no less than 60%. Hence, an appropriate amount of pollen was spread on white paper, which was subsequently placed in a glass petri dish. This dish was then incubated in an artificial climate box under conditions of 25 °C, 95% relative humidity, and darkness. Following the set incubation period, the activated pollen was used to prepare the pollen suspension as per the given formulation [26]. The germination rate of the pollen was then assessed following the protocol outlined

in Section 2.1. The experiment was conducted indoors on 2 April 2022, at an ambient temperature of 18 °C and 45% relative humidity. Each experimental set was replicated three times, with each replicate consisting of five samples. The study aimed to investigate the influence of incubation times ranging from 0 h to 20 h on the pollen germination rate. Given that each experimental group was separated by 2 h intervals, a total of 11 sets were examined.

### 2.3. Experiment on the Effect of Pollen Suspension Temperature on the Pollen Germination Rate

To simulate the impact of pollen suspension temperature within the tank on the pollen germination rate, equipment capable of maintaining specific temperatures for the pollen suspension was necessary. A digital thermometer (range −20 °C~60 °C, accuracy ± 0.5 °C, WSB-5-H2, YipinBoyang Technology Co., Ltd., Shenzhen, China), designed for water temperature measurement, and a water bath (range 5 °C~100 °C, accuracy ± 1.0 °C, HH2, Bona Technology Co., Ltd., Hangzhou, China), ensuring a consistent temperature for the pollen suspension, were chosen. After preparation, the pollen suspension was transferred into centrifuge tubes and placed in the water bath for 20 min (Figure 2). It is imperative to note that the temperature set in the water bath must align with that of the prepared pollen suspension. On 5 April 2022, the experiment was conducted indoors with an ambient temperature of 18 °C and 44% relative humidity. For this study, pollen suspension temperatures ranged from 10 °C to 45 °C, with each treatment group having a 5 °C temperature difference. An ambient temperature of 18 °C served as the control group (CK). Consequently, a total of nine experimental groups were set up, each replicated three times. For each test, five samples were collected.

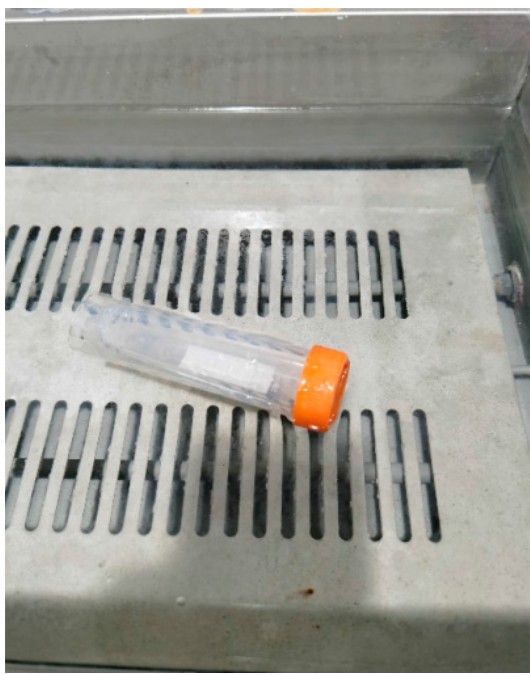

**Figure 2.** Place the centrifuge tube containing the pollen suspension into the water bath. Once positioned, secure the water bath by closing its lid.

### 2.4. Experiment on the Effect of Pollen Suspension (in the Tank) Pre-Germination Temperature on Fruit Set and Fruit Size

In the present study, pear trees that were naturally pollinated were designated as the CK. Ten groups were arranged for liquid spray pollination tests, based on the pollen suspension temperatures configured as per Section 2.3, with each test being replicated three times. On 8 April 2022, thirty rows of six-year-old, unbloomed pear trees were selected in the Fojianxi Pear Garden, Xiying Village, Yukou Town, Pinggu District, Beijing (40.1962° N,

116.9902° E). Within each row, three pear trees were randomly chosen. For every test fruit tree, fifteen test inflorescences were utilized to compile relevant experimental data. To mitigate the impact of wind and insects on pollination outcomes, these inflorescences were encased in breathable non-woven bags (Figure 3a), and red plastic tags were affixed to the branches of the test inflorescence for identification.

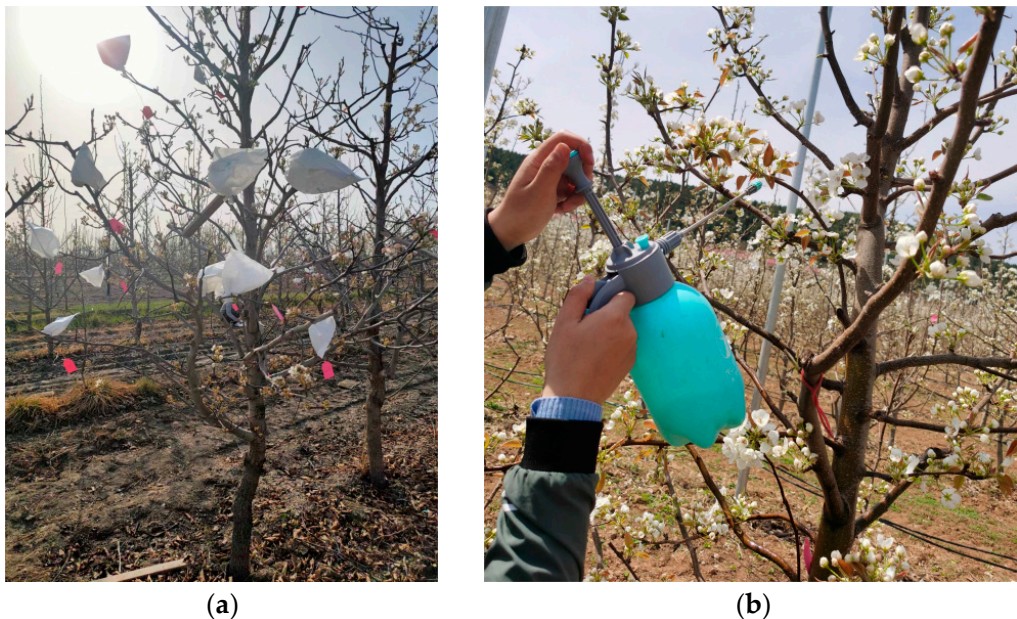

**(a)** **(b)**

**Figure 3.** Preparation and spraying method for orchard liquid spray pollen test. (**a**) Pear inflorescences were enclosed in non-woven bags, each identified by a red plastic tag. (**b**) A hand-held manual air pressure sprayer was utilized to administer the pollen suspension, maintaining consistent pressure throughout to ensure stable spray pressure.

The liquid spray pollination experiment was executed on 13 April 2022. Throughout the experiment, weather conditions were sunny, with temperatures ranging from 22 to 25 °C, a relative humidity of 42%, and wind speeds fluctuating between 0.8 and 1.5 m/s. Non-woven bags were removed 0.5 h before the experiment. Pear flowers with stigmas (have no stigmatic receptivity), or those in an unbloomed state, were excised, and the number of remaining inflorescences and flowers was noted on a red plastic tag. Pollen suspensions at varying temperatures were sprayed using a handheld manual air pressure spray can (Figure 3b), which was selected due to its minimal impact on pollen activity during the spraying of the pollen suspension [26]. Each inflorescence was sprayed for a duration of 2–3 s. Post-spraying, the targeted inflorescences were re-encased in non-woven bags. It is pertinent to note that non-woven bags were not removed from the naturally pollinated pear trees throughout the experiment.

The non-woven bags were removed on 23 June 2022. Pertinent information about the flowers and inflorescences, recorded on the red plastic tags, along with the number of set pear fruits, was documented in a notebook (Figure 4a). These data were subsequently employed to calculate the fruit set rate of flowers and inflorescences. On 20 September 2022, the longitudinal and transverse dimensions of Fojianxi pears were measured thrice using a slide gauge (Figure 4b; Model: DL91150, range: 0~150 mm, accuracy: ±0.03 mm, Deli Technology Co., Ltd., Ningbo, China). All dimension measurements of the Fojianxi pears were meticulously recorded in a notebook for ensuing statistical analysis.

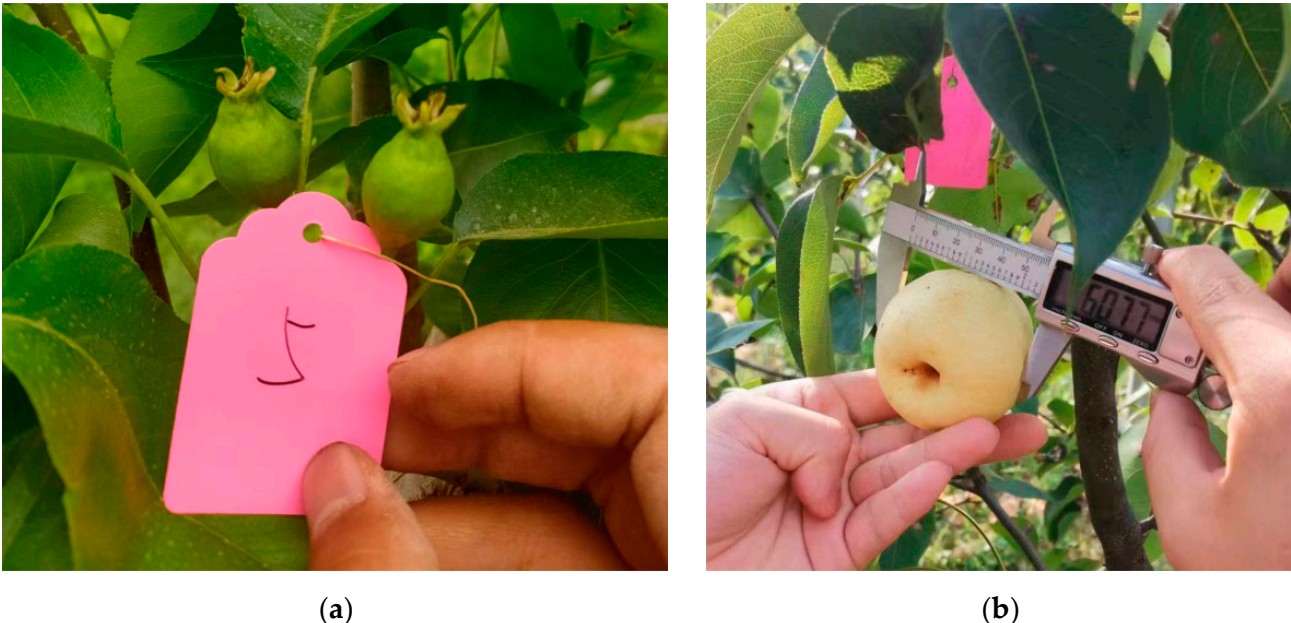

(**a**)　　　　　　　　　　　　　　(**b**)

**Figure 4.** Statistics on fruit set and fruit size. (**a**) Documentation of flower and inflorescence fruit set. (**b**) Evaluation of longitudinal and transverse dimensions of Fojianxi pears. The red plastic tags, arranged in accordance with branch development, contain detailed information regarding the flowers and inflorescences. The number of inflorescences corresponds to the data specified on the label. It is imperative to accurately locate the fruit's most substantial transverse stem and perform multiple measurements utilizing a slide gauge.

*2.5. Data Processing*

In this study, the pollen germination rate was calculated using the formula given in Equation (1). A larger percentage signifies a heightened pollen germination rate, and for all experimental sets, the percentage was determined to reflect the germination rate.

$$G_r = \left( \frac{T_g}{T_p} \right) \times 100\% \tag{1}$$

where $G_r$ represents pollen germination rate, $T_g$ denotes the number of germinated pollen grains, and $T_p$ signifies the total number of pollen grains observed within the field of view.

The fruit set rate of flowers, defined in Equation (2), signifies the likelihood of an individual flower developing into fruit.

$$F_r = \left( \frac{N_f}{N_t} \right) \times 100\% \tag{2}$$

where $F_r$ represents the fruit set rate of flowers; $N_f$ denotes the number of successful fruit settings; $N_t$ symbolizes the total number of flowers involved in the liquid spray pollination test.

Similarly, the fruit set rate of inflorescences, articulated in Equation (3), represents the probability of an inflorescence yielding fruit.

$$I_r = \left( \frac{\sum_{i=1}^{n} if(N_i)}{N_s} \right) \times 100\% \tag{3}$$

where $I_r$ signifies the fruit set rate of inflorescences; $N_i$ denotes the fruit count of an inflorescence; $if$ indicates whether an inflorescence has borne fruit (output 1) or not (output 0); $N_s$ denotes the total number of inflorescences examined.

Upon fruit ripening, the roundness of pears can be quantified using the fruit shape index. As delineated in Equation (4), a fruit shape index closer to one indicates a rounder pear.

$$F_i = \left( \frac{L_p}{H_p} \right) \times 100\% \tag{4}$$

where $F_i$ represents the fruit shape index; $L_p$ represents the transverse dimension of fruit; $H_p$ represents the longitudinal dimension of fruit.

The Coefficient of Variation ($CV$) serves as an indicator of data dispersion across different measurement groups, calculable via Equation (5).

$$CV = \left( \frac{SD}{MN} \right) \times 100\% \tag{5}$$

where $SD$ represents the standard deviation of the group data, and $MN$ represents the mean of the group data.

$MN$ and $SD$ can be expressed by Equations (6) and (7), respectively.

$$MN = \frac{\sum_{i=1}^{n} P_i}{n} \tag{6}$$

where $n$ denotes the sample count and $P_i$ denotes the $i$th sample's data.

$$SD = \sqrt{\frac{\sum_{i=1}^{n} (P_i - MN)^2}{n}} \tag{7}$$

The experimental data were graphed using OriginPro Version 2020 (OriginLab Inc., Northampton, MA, USA). Prior to any advanced data analysis, the data underwent a normality test using SPSS Statistics Version 22 (IBM Inc., Armonk, NY, USA). Further, the data was subjected to a one-way analysis of variance (ANOVA) utilizing SPSS. Duncan's multiple range test, a post hoc test within SPSS's one-way ANOVA, was also executed. In the context of this study, a *p*-value of 0.05 was considered indicative of a significant difference. All data adhered to a normal distribution.

## 3. Results

### 3.1. Results of Incubation Duration on Pollen Germination Rate

The vitality of pollen grains exhibited an initial rise followed by a decline as the duration of incubation increased (Figure 5). The vitality was at its nadir at 0 h, registering at $6.72 \pm 0.52\%$. With the elongation of incubation duration, the pollen germination rate surged swiftly, peaking at 6 h with a value of $75.89 \pm 8.14\%$. As the incubation persisted beyond this point, there was a gradual decline in the vitality of pollen grains, reaching $10.51 \pm 1.04\%$ at 20 h. Interestingly, the CV for pollen germination rate followed a trajectory of initial decrease, subsequent increase, and a final decline as time advanced. The minimum CV for pollen germination was 5.32% at 2 h, and it spiked to its maximum of 34.15% at 12 h. Concurrently, after eliminating anomalies, we conducted an ANOVA analysis using SPSS. The analytical outcomes revealed a pronounced and significant difference in pollen germination rates across various incubation durations ($p < 0.001$, Table 2). The germination rate at 6 h was notably superior to other time intervals. Similarly, the rate at 8 h was significantly elevated compared to all other intervals except for 6 h. Contrarily, the disparities among other incubation durations were not significant.

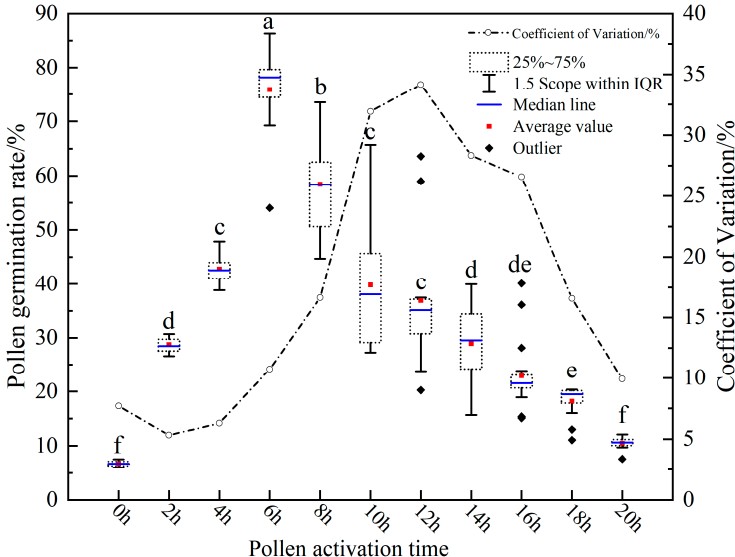

**Figure 5.** Results of pollen incubation duration on pollen germination rates. Different letters indicate significant differences (Duncan test, $\alpha = 0.05$).

**Table 2.** ANOVA results of between group for different experiments. The data in the table were obtained from SPSS.

| Sources of Variation for Different Experiments | Sum of Squares | DF | Mean Squares | F | Sig. |
|---|---|---|---|---|---|
| Pollen activation time and pollen germination rate | 54,635.567 | 10 | 5463.557 | 107.676 | 0.000 *** |
| Pollen suspension temperature (PST) and pollen germination rate | 45,095.424 | 8 | 5636.928 | 186.270 | 0.000 *** |
| PST and fruit set rate of flowers | 17,717.643 | 9 | 1968.627 | 410.933 | 0.000 *** |
| PST and fruit set rate of inflorescences | 40,174.583 | 9 | 4463.843 | 362.652 | 0.000 *** |
| PST and fruit longitudinal dimension | 40,174.583 | 9 | 4463.843 | 362.652 | 0.000 *** |
| PST and fruit transverse dimension | 3074.531 | 9 | 341.615 | 10.741 | 0.000 *** |
| PST and fruit shape index | 906.996 | 9 | 100.777 | 1.238 | 0.290 |

Statistical significance level: *** $p < 0.001$.

### 3.2. Effect of Suspension (in the Tank) Temperature on the Germination Rate of Pollen

At lower pollen suspension temperatures ($\leq 20\ °C$, Figure 6a–c,i), there was a preponderance of ungerminated pollen grains. As the temperature incrementally rose to ($\leq 35\ °C$, Figure 6d,e), the count of ungerminated pollen grains markedly diminished. When the temperature escalated further to $40\ °C$, the pollen grains were predominantly ungerminated (Figure 6g). This underscores that the optimal suspension temperature bracket conducive for pollen grains ranges between $20\ °C$ and $35\ °C$. Beyond the threshold of $40\ °C$, the pollen grains succumbed (Figure 6h).

The observed patterns in the aforementioned statistics mirrored those seen in Figure 7a. Specifically, as the pollen suspension temperature increased, the pollen germination rate initially rose slowly and then plummeted. At a pollen suspension temperature of $10\ °C$, the germination rate was a mere $21.13 \pm 4.01\%$. The zenith of $72.06 \pm 6.87\%$ was attained at $30\ °C$. Beyond this, a sharp decline was noted at $40\ °C$, where the rate plunged from $61.11 \pm 4.27\%$ at $35\ °C$ to $7.14 \pm 3.33\%$ at $40\ °C$. At an elevated $45\ °C$, the germination rate dwindled to $2.62 \pm 1.65\%$. The CV in pollen germination displayed a dynamic pattern: it initially decreased, then increased, followed by another decrease, and finally surged. The $35\ °C$ temperature marked the lowest CV at 11.7%, while the $45\ °C$ registered the highest at 59.29%. Additionally, the CV at $35\ °C$ was notably expansive, approximating 44%.

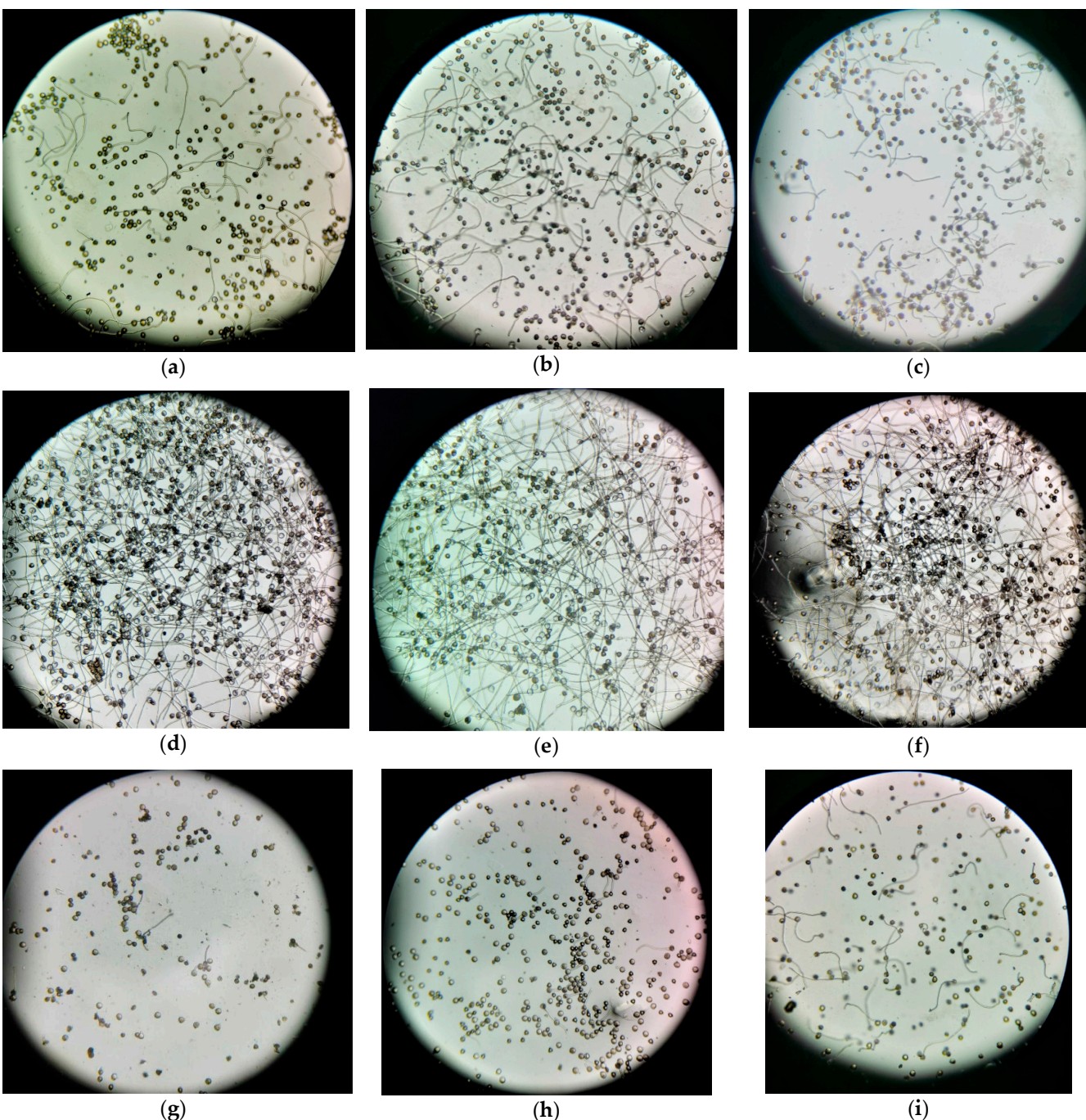

**Figure 6.** Pollen grain germination at various pollen suspension temperatures was captured using a Xiaomi 12X mobile phone. Panels (**a**–**i**) represent temperatures of 10 °C, 15 °C, 20 °C, 25 °C, 30 °C, 35 °C, 40 °C, 45 °C, and 18 °C, respectively. The designated temperature represents both the pollen suspension and the water bath set temperatures. Germination rates in the selected fields of view from Figure 6a–i were 21.26%, 21.23%, 38.22%, 64.35%, 71.26%, 62.02%, 7.26%, and 27.05%, respectively, with clear visual delineation of pollen grain germination.

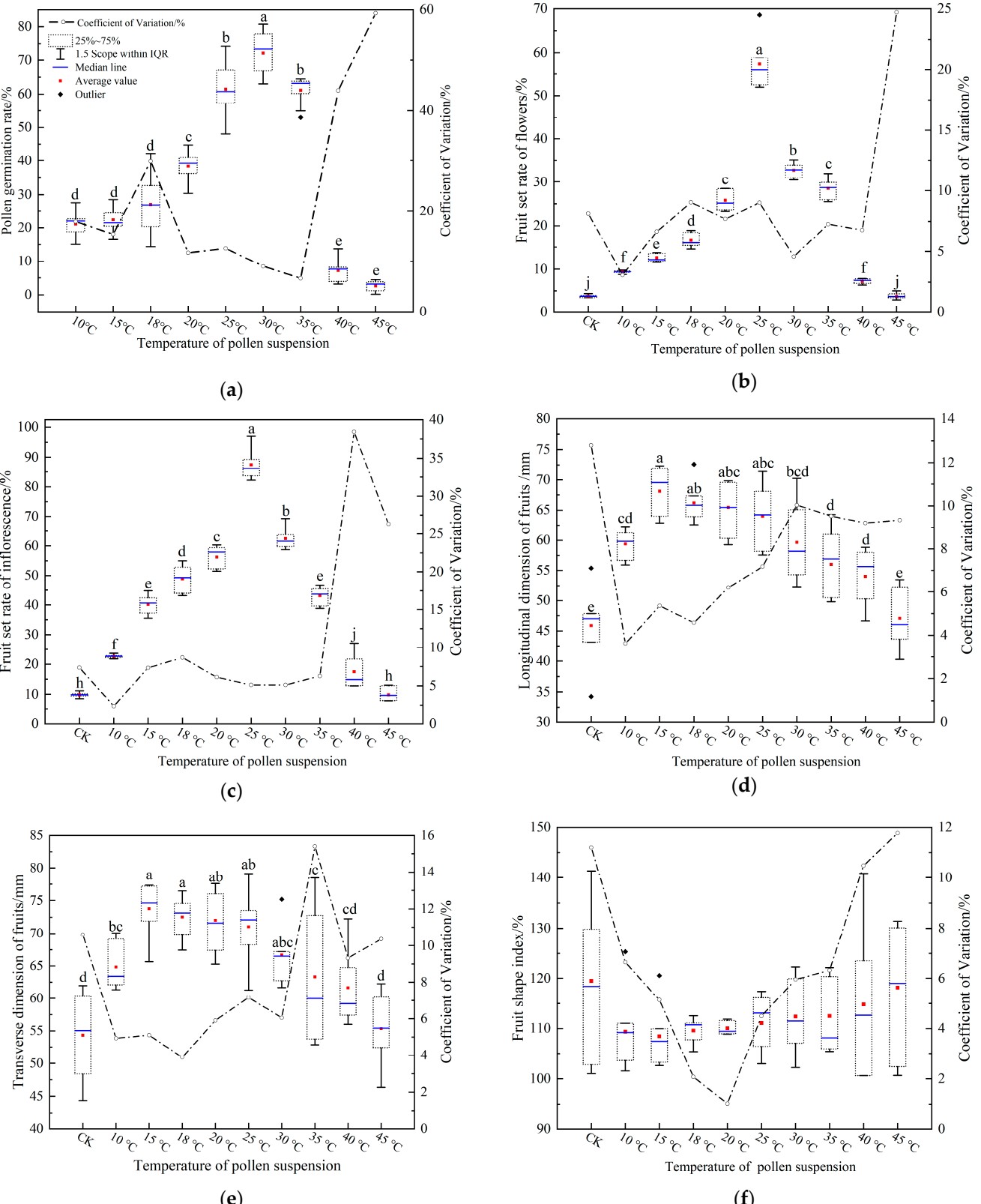

**Figure 7.** Results of pollen suspension temperature on the pollen germination rate, fruit set and fruit size. (**a**) pollen germination rate, (**b**) fruit set rate of flowers, (**c**) fruit set rate of inflorescences, (**d**) fruit longitudinal dimension, (**e**) fruit transverse dimension, and (**f**) fruit shape index. Legends in Figure 7b–f correspond with those in Figure 7a. Items in Figure 7i do not exhibit significant differences from one another, with different letters indicating significant differences (Duncan test, α = 0.05).

Upon analysis of the data using SPSS, a highly significant difference was observed between pollen suspension temperature and pollen germination rate ($p < 0.001$; Table 2). Three distinct temperature zones were identified: low ($\leq 20$ °C), middle ($\leq 35$ °C), and high ($>35$ °C). These results concur robustly with our visual observations.

In the low-temperature zone, pollen germination at a suspension temperature of 20 °C ($38.33 \pm 4.76\%$) was significantly elevated compared to 15 °C ($22.37 \pm 3.64\%$) and 18 °C ($27.05 \pm 9.16\%$). Within the medium temperature zone, the 30 °C pollen suspension exhibited a notably higher germination rate compared to other temperatures, while no significant difference was detected between germination rates at 25 °C ($61.47 \pm 8.20\%$) and 35 °C ($61.11 \pm 4.27\%$). In the high-temperature zone, the pollen germination rate was markedly lower. This suggests that optimal pollen germination rates are sustained at pollen suspension temperatures ranging from 20 to 35 °C.

### 3.3. Effect of Pollen Suspension Temperature on Fruit Set

The fruit set of flower and inflorescence demonstrated a trend of increasing and subsequently decreasing with the pollen suspension temperature (Figure 7b,c). There was a highly significant difference ($p < 0.001$, Table 2) between the pollen suspension temperature and both of them. The pinnacle for both metrics was observed at a pollen suspension temperature of 25 °C, registering $57.29 \pm 5.58\%$ and $87.50 \pm 4.84\%$ respectively. When the pollen suspension temperature reached 45 °C, the flower and inflorescence fruit set rates ($3.75 \pm 0.73\%$ and $9.87 \pm 0.78\%$) were not statistically comparable to the CK ($3.75 \pm 0.33\%$ and $9.87 \pm 2.25\%$). Nonetheless, significant disparities were identified between the other pollen suspension temperatures and the CK. This indicates that variations in fruit set and pollen germination may be caused by pollen suspension temperature. Fruit set rates of flowers at pollen suspension temperatures of 20 °C ($25.79 \pm 2.14\%$) and 35 °C ($28.55 \pm 2.25\%$) were statistically indistinguishable. However, a marked difference was discerned for inflorescence fruit set rates ($56.25 \pm 3.71\%$ vs. $43.17 \pm 2.92\%$). Pollen suspensions at 35 °C had higher pollen germination rates, which was significantly higher than 20 °C and 15 °C (Figure 7a). After the fruit set, the fruit set rate of flowers was equivalent only to the 20 °C temperature, while for inflorescence, it was only akin to 18 °C. These findings suggest disparities between pollen germination rates and fruit set outcomes as a function of pollen suspension temperature.

The highest pollen germination rate was observed at a pollen suspension temperature of 30 °C, while the peak fruit set rate was identified at 25 °C. Notably, pollen suspensions exhibiting robust germination at elevated temperatures (30 °C and 35 °C) did not demonstrate a disparity in fruit set when compared to suspensions with diminished germination at reduced temperatures (15~20 °C). CVs for both flower and inflorescence fruit set rates remained below 10%, with the exception of pollen suspensions at 45 °C.

### 3.4. Effect of Pollen Suspension Temperature on Fruit Size

As the pollen suspension temperature increased from 10 °C to 45 °C, both the longitudinal and transverse dimensions of the 'Fojianxi' pear fruit exhibited an initial increase followed by a subsequent decrease(Figure 7d,e). These dimensions displayed pronounced differences in relation to the pollen suspension temperature, with significance levels below $p < 0.001$ (Table 1). The CVs for both parameters were under 16%. At a temperature of 15 °C, the peak values for fruit longitudinal and transverse dimensions were recorded as $68.12 \pm 3.94$ mm and $73.77 \pm 4.04$ mm, respectively. When compared with the CK, the fruit dimensions in the context of the pollen suspension were significantly augmented across all temperatures, barring the exception at 45 °C. The fruit shape index displayed an initial decline followed by an increase as the temperature of the pollen suspension rose. No significant difference was observed between the fruit shape index and the pollen suspension temperature ($p = 0.290 > 0.05$, Table 2). The CV for fruit shape index surpassed 10% for both the CK and the 45 °C pollen suspension, while all other temperatures remained below this threshold. Specifically, the fruit shape index was the smallest ($108.42 \pm 6.02\%$)

at 15 °C, whereas the CK registered the highest value (119.51 ± 14.46%). In summary, the longitudinal and transverse dimensions of the fruit, as well as the fruit shape index, exhibited optimal results at a suspension temperature of 15 °C.

## 4. Discussion

This study uncovered an intriguing observation: the optimal suspension temperatures for optimal pollen germination rate (Figure 7a), fruit set effects (flower and inflorescence, Figure 7b,c), and fruit shapes (longitudinal and transverse dimensions of fruit and fruit shape index, Figure 7d–f) were not consistent. Specifically, the maximal pollen germination rate was noted at 30 °C (Figure 7a), and the peak fruit set rate via liquid spray pollination occurred at 25 °C (Figure 7b,c). Additionally, the optimal fruit shapes (longitudinal and transverse dimensions of fruit and fruit shape index) following liquid spray pollination were achieved at 15 °C (Figure 7d–f).

It was shown that *Pyrus bretschneiderilia* had the highest pollen germination rate of 66.3% at an in vitro culture temperature of 25 °C [24,27]. In this study, we found that the pollen germination rate was 61.47 ± 8.20% when both the pollen suspension and in vitro culture temperature was 25 °C. This was not significantly different from the data of the research. However, the pollen germination rate was 72.06 ± 6.87% when the pollen suspension temperature was 30 °C (25 °C for in vitro culture in this study). There was a larger increase in pollen germination rate over the abovementioned study. There was evidence of complete or almost complete inhibition of pollen tube growth at low (10 °C) and high (35 °C) temperatures [17]. As far as pollen germination rate is concerned, 35 °C in vitro incubation temperature had a higher pollen germination rate than 10 °C. This is consistent with our findings. Moreover, the length and width of pollen tubes were higher at 10 °C in vitro incubation temperature than at 35 °C. These data were not measured in this study.

We investigated the effect of short-term (less than 20 min) pollen suspension temperature on pollen germination rate. Then pollen suspensions were incubated in vitro for a long time (3 h) at a temperature of 25 °C. This, however, caused a significant difference in germination rate. It has been shown that low temperature (4 °C) inhibits oxygen consumption and ATP production in pollen grains [28]. High temperatures disrupt the structure of enzymes in pollen [29]. However, all of the abovementioned studies were conducted with long durations of in vitro culture. In the study, the in vitro incubation temperature was optimum. Similarly, it has been shown that high- or low-temperature stress needs to be sustained for a longer duration. Additionally, alternating high/low temperatures had a lower inhibitory effect on pollen germination than continuous high or low temperatures [30]. At higher temperatures ($\leq$35 °C), the enzymes in pollen grains are more active for a short period of time (less than 1.5 h), which will enhance the pollen germination rate but will consume too much energy from pollen [29]. It is the reason for the higher pollen germination rate observed in higher temperature pollen suspensions within 3 h [31]. Lower temperatures (10 °C) inhibited pollen germination, but the inhibition was slight. Moreover, lower temperatures enhance pollen germination energy [32]. A short period of higher or lower temperature stress is not very inhibitory to pollen germination rate [30]. What is changed in this study is the temperature of pollen suspension, which is placed into a culture and takes some time to reach ambient temperature. This can be seen as alternating low/high temperatures; however, the pollen takes a long time (>3 h) to resume pollen tube growth from low temperature to optimum temperature [33]. In addition, the low-temperature pollen suspension needs to be slowly raised to ambient temperature. This is the reason for the lower pollen germination rate observed in the low-temperature pollen suspensions within 3 h.

The phenomena of fruit set (flower and inflorescence) and fruit shapes (longitudinal and transverse dimension of fruit and fruit shape index) not only relate to the activity of pollen but are also intricately connected to the ultimate elongation and width of the pollen tubes. Previous researchers have ventured into this domain, defining "germination energy"

as an amalgamation of the initiation time of pollen germination and the ultimate length and width of pollen tubes [32]. Lower temperatures have been described above to enhance pollen germination energy. Furthermore, pear pollination has a wide temperature range to resist drastic temperature changes [34]. The appropriate temperature for pear pollen and stigmas is highly dependent on the temperature of their growth region [35]. They are adapted to local temperature variations. Pear flowers in temperate zones have pollen tubes that grow no less rapidly at lower temperatures than at the appropriate temperature, and lower temperatures are suitable for stigmatic receptivity. Whereas the Fojianxi pear is native to Beijing, the *Pyrus bretschneiderilia* can survive in Beijing. At 10 °C, all adherent pollen could germinate and penetrate the disseminating ovules, while at 20 °C they recorded 90% germination and penetration, and at 30 °C only 27% germination and 22% penetration [36]. Moreover, they found no significant effect of temperature on the adhesive effect of stigma. However, low temperature increased the time of stigma receptivity to pollen to 9 days (2 days at 30 °C). This means that the pollen inside the stigma had a longer time to grow at low temperatures. Meanwhile, pollen germination was delayed by returning to the appropriate temperature after low-temperature stress, and there was no significant effect on pollen germination rate [35,36]. However, high pollen suspension temperatures (40 °C and above) not only caused damage to pollen grains but can also damage pollen stigmas. Low pollen suspension temperatures would reduce the temperature around the stigma for a short time and elevate the penetration. This may be one of the reasons for the elevated fruit set. At higher temperatures (30 °C and 35 °C), the pollen suspension briefly raises the temperature around the stigma and reduces penetration. This is likely to be one of the reasons for the reduced fruit set.

While previous studies have simulated field temperatures using in vitro cultures, this approach differs from our method, which focused on suspension temperature. In our field experiments, the process of preparing the pollen suspension and spraying it onto the stigma took under 20 min. During this period, the pear pollen did not initiate germination. We postulate that the pollen grains were accumulating germination potential. Ambient temperatures during our experiments ranged between 22 °C and 25 °C. Over the subsequent five days, temperatures fluctuated between 8 °C and 22 °C without precipitation, providing favorable conditions for pollen germination. Yet, the outcomes varied, highlighting the significant influence of brief exposure to pollen suspension temperature on its germination potential. This effect was evident in the highest fruit set rate observed at a 25 °C pollen suspension, attributed to superior germination rates and energy at this temperature—a feature absent at 30 °C. Conversely, the most potent germination energy was observed at 15 °C, yielding the best fruit morphology. Extremely low suspension temperatures, however, diminished this germination potential. In addition, the temperature of pollen suspension had a temporal effect on the stigma capacity. In addition to irreversible effects, the stigma is mainly affected by ambient temperature. Therefore, pollen germination energy is the main cause of pear fruit set and size.

Some research has explored fruit morphology [37]. A significant positive correlation has been identified between fruit size and seed quantity [38]. A more robust relationship was found between seed count and the fruit's longitudinal dimension than its transverse dimension [39]. The ultimate length of the pollen tube is pivotal for successful pollination. Elevated pollen suspension temperatures, which decrease germination energy, have been negatively correlated with the fruit's longitudinal axis (as depicted in Figure 7d). Fruit morphology also positively correlates with seed shape [40]. We theorize that excessively high pollen suspension temperatures can alter pollen's germination energy, leading to seed deformities, and subsequently affecting the fojianxi pear's dimensions. This theory explains the negative correlation observed between suspension temperature and the fruit's longitudinal dimension. Concerning the pollen activation duration, we speculate that overly extended periods (>6 h) cause pollen to absorb water, swell, and ultimately perish, while exceedingly brief durations inadequately activate the pollen. This hypothesis requires further experimental validation.

In summary, elevated pollen suspension temperatures decrease the accumulated germination potential of pollen grains, leading to high germination rates but shorter final pollen tube lengths. This can damage the stigma or reduce stigmatic capacity. Lower suspension temperatures enhance the cumulative germination potential, resulting in reduced germination rates but extended final pollen tube lengths. This can improve stigmatic capacity. Excessively high temperatures can be lethal to pollen grains, while extremely low temperatures reduce their germination energy. We recommend a pollen suspension temperature range of 15 °C to 25 °C for spraying, with an optimal temperature of 25 °C for this study.

## 5. Conclusions

In the present study, pollen germination rates were assessed by manipulating the pollen activation time and altering the temperature of the pollen suspension. Different temperatures of pollen suspensions were sprayed onto stigmas to acquire data regarding the fruit set rate of flowers and inflorescences, as well as the longitudinal and transverse dimensions of fruits and fruit shape indices under varying temperatures. The subsequent experiments led to the following conclusions:

(1) The ideal pollen activation time was determined to be 6 h. Prolonged activation could induce pollen swelling and subsequent death due to water absorption. Although the highest pollen germination rate was observed at a pollen suspension temperature of 30 °C, the peak fruit set rates for both flowers and inflorescences were achieved at a pollen suspension temperature of 25 °C. The optimal longitudinal and transverse dimensions of fruits, as well as the fruit shape index, were observed when the pollen suspension temperature was maintained at 15 °C.

(2) Lower pollen suspension temperatures (15~25 °C) exhibited elevated germination energy. Conversely, higher pollen suspension temperatures (30~35 °C) demonstrated diminished germination energy. Despite the higher germination rates exhibited by the latter, their flower and inflorescence fruit set rates were merely equivalent to those at lower temperatures (15~20 °C).

(3) The discrepancies in germination energy resulted in enhanced fruit morphology following spraying at lower pollen suspension temperatures, notably reaching an optimum at 15 °C, indicating peak germination energy at this temperature. The potential cause of this phenomenon could be attributed to variations in seed morphology following pollination with pollen of differing germination energies.

While intriguing conclusions have been drawn, certain aspects remain unresolved. Because the short observation time (3 h) may result in the observation of the germination rate may be incomplete, and a long incubation period (12 h) is needed for future observation. Due to negotiations with farmers, the experimental Fojianxi pears were not dissected to examine their internal seeds. Additionally, the proposed relationship between germination energy and fruit seed shape discussed in the previous sections necessitates further exploration. The temperature of the pollen suspension for a short time affects pollen germination, germination energy, and stigmatic capacity. Therefore, we recommend farmers measure the water temperature when preparing pollen suspension. This is because proper pollen suspension temperature will improve the fruit set. We also hope that relevant researchers will participate in the study of the effect of the temperature of pollen suspension on pollen germination and fruit set in the tank.

**Author Contributions:** Conceptualization, L.L., X.H. and Y.L. (Yajia Liu); methodology, L.L. and Z.L.; software, L.L. and Z.L.; validation, X.H. and Y.L. (Yajia Liu); formal analysis, L.L., X.H. and Y.L. (Yajia Liu); investigation, L.L., Z.L., H.H., Y.L. (Yangfan Li) and B.Q.; resources, X.H.; data curation, L.L., Z.L., H.H., Y.L. (Yangfan Li) and B.Q.; writing—original draft, L.L., Y.L. (Yangfan Li), X.H. and Y.L. (Yajia Liu); visualization, L.L. and H.H.; and funding acquisition, X.H. All authors have read and agreed to the published version of the manuscript.

**Funding:** The earmarked fund for CARS: CARS-28, Sanya Institute of China Agricultural University Guiding Fund Project (Grant No. SYND-2021-06), the 2115 Talent Development Program of China Agricultural University.

**Institutional Review Board Statement:** Not applicable.

**Informed Consent Statement:** Not applicable.

**Data Availability Statement:** The data presented in this study are available on request from the corresponding author.

**Acknowledgments:** The authors would like to give special thanks to Wang Zhong for providing the test orchard.

**Conflicts of Interest:** The authors declare no conflict of interest.

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
