# Peer review of "Effect of Pre-Germination Temperature Regime on Pollen Germination and Fruit Set in Pear, Pyrus bretschneiderilia"

_horticulturae, doi:10.3390/horticulturae9101151_

Round 1

Reviewer 1 Report

It would be important to present the results of the F tests applied clearly (I suggest a table). This may have implications for the analyzes presented regarding the media test used.

Figures 05 and 07 of the results present a lot of information, therefore, I suggest improving the resolution and avoiding strong colors.

The correlation analysis showed significant results even with very low values ​​(i.e: -0.25). Was the test carried out with average data from treatments/replications or were all observations used?

In the conclusions, with excess of the last paragraph, the authors only summarized the results. Please present impacts and future perspectives, if possible.

Author Response

Thank you very much for your comments, we have made the changes. Our response is attached. We have uploaded the revised manuscript.

Reviewer 2 Report

Dear author, please have the language and writing corrected. The data are fine just the text is not understandable. Examples are in the attached word file.

see file

Author Response

(The authors gave the same response as above.)

Round 2

Reviewer 2 Report

Dear authors, thank you for answering my questions.